# Hereditary Hemorrhagic Telangiectasia in Pediatric Age: Focus on Genetics and Diagnosis

**Cesare Danesino, Claudia Cantarini \* and Carla Olivieri**

General Biology and Medical Genetics Unit, Department of Molecular Medicine, University of Pavia, 27100 Pavia, Italy

\* Correspondence: claudia.cantarini01@universitadipavia.it

**Abstract:** Hereditary Hemorrhagic Telangiectasia (HHT) or Rendu–Osler–Weber Syndrome (ROW) is an autosomal dominant vascular disease, with an estimated prevalence of 1:5000. Genes associated with HHT are *ACVRL1*, *ENG*, *SMAD4*, and *GDF2*, all encoding for proteins involved in the TGFβ/BMPs signaling pathway. The clinical diagnosis of HHT is made according to the "Curaçao Criteria," based on the main features of the disease: recurrent and spontaneous epistaxis, mucocutaneous telangiectases, arteriovenous malformations in the lungs, liver, and brain, and familiarity. Since the clinical signs of HHT can be misinterpreted, and the primary symptom of HHT, epistaxis, is common in the general population, the disease is underdiagnosed. Although HHT exhibits a complete penetrance after the age of 40, young subjects may also present symptoms of the disease and are at risk of severe complications. Here we review the literature reporting data from clinical, diagnostic, and molecular studies on the HHT pediatric population.

**Keywords:** hereditary hemorrhagic telangiectasia; HHT; pediatric age; rare disease

## 1. Introduction

Hereditary Hemorrhagic Telangiectasia (HHT), formerly known as Rendu–Osler–Weber disease, is an autosomal dominant vascular dysplasia.

Historically, credit is given to Henry G. Sutton [1] for his description in 1864 of a disorder characterized by epistaxis and degeneration of the vascular system, likely the first report of the disease.

Shortly thereafter, new observations were added to Sutton's. Benjamin Guy Babington noted that, in some families, nosebleeds could be an inherited trait [2]; Henry Rendu reported on "petits angiomes cutanes et muqueux" [3]; Sir William Osler presented a family showing the association of recurrent epistaxis with skin and/or mucous telangiectases [4]; and, in 1907, Frederick Parkes Weber published his description of multiple hereditary angiomata associated with recurrent hemorrhage [5]. The name "hereditary hemorrhagic telangiectasia" was finally coined by Hanes [6].

Since then, a large amount of clinical epidemiological and genetic data has been produced, extensively reviewed by Jamie McDonald and David A. Stevenson [7].

## 2. Epidemiology

HHT affects at least one million people worldwide, but its true prevalence is difficult to evaluate, as the full clinical picture may be evident only in adulthood, and the disease is often underdiagnosed [8]. In fact, when a new patient is diagnosed, it is common to experience the subsequent identification of previously undiagnosed relatives. In addition, after the first report of a founder effect in the Netherland Antilles [9], the same observation has been reported in many different countries, including Italy [10]. Overall, a reasonable general estimate of the prevalence is 1 in 5000 [11].

### 3. Genetics, Molecular Genetics and Pathogenesis

HHT is classified as an autosomal dominant disorder, and data from formal genetics have been confirmed by the demonstration of heterozygous mutations in three main genes: *ACVRL1*, *ENG*, and *SMAD4*. In the last decade, *GDF2* was reported to be causative of HHT in fewer than 20 cases.

Very large pedigrees spanning over many generations and including tens of cases are known in each HHT center, and their presence suggests that fitness of patients is not (severely) reduced, although no detailed studies are available; penetrance increases during the lifetime, and, in adulthood, over the age of 40, it is estimated to be above 95% [8].

Endoglin (*ENG*) (9q34.11, OMIM 187300) was the first disease-related gene to be identified in 1994 [12], followed two years later by the demonstration of pathogenic mutations in the activin A receptor type II—like I (*ACVRL1*) (12q13.13, OMIM 601284) [13].

Over 95% of patients carry pathogenic mutations (missense, splice site, deletions, duplications) in one of the two above-mentioned genes, and HHT1 nowadays refers to patients carrying mutations in *ENG,* while variants in *ACVRL1* defines HHT2 [14]. All details (type of mutation, references, year of reporting) are entered in the HHT Database, updated on March 2021 [15]. Up to now, the pathogenic mutations reported in each gene are mutually exclusive: in each patient, if a germinal pathogenic mutation is found, for instance in *ACVRL1*, no other germinal pathogenic mutation will be found in any of the other disease-related genes [16].

One single case is reported in which a germinal mosaicism involving *ACVRL1* was found in the same patient. Each pathogenic variant was transmitted independently to two different family branches [17].

A small number of cases (approximately 2%) shows a complex clinical picture, including the clinical signs of Juvenile Polyposis Syndrome (JPS) and HHT, and carries mutations in the *SMAD4* gene (18q21.2, OMIM 600993) [18]. Likely, an additional HHT subtype is related to mutations in the *GDF2* gene (10q11.22, OMIM 605120) [19]. The existence of cases with an HHT-like phenotype but no pathogenic mutation in the previously quoted genes underlines the need for extensive molecular investigation, as a broad genetic heterogenicity is likely to exist. Details about other genes whose mutations cause phenotypes possibly related to or partially overlapping HHT, are beyond the scope of this review and can be found in [7].

There is a general agreement stating that HHT originates from loss of function of proteins coded for by genes of the transforming growth factor beta (TGF-β) signaling pathway (Figure 1): *ENG* encodes endoglin, a type III receptor mainly expressed in endothelial cells; *ACVRL1* encodes a serine/threonine receptor kinase R3, ALK1, which is a type I cell-surface receptor for the TGF-superfamily of ligands, and it is predominantly expressed in endothelial cells as well [20]; *SMAD4* encodes a protein acting as the nuclear signaling effector in the same pathway; and *GDF2* encodes the secreted ligand BMP9, which represents the physiological ligand for ALK1.

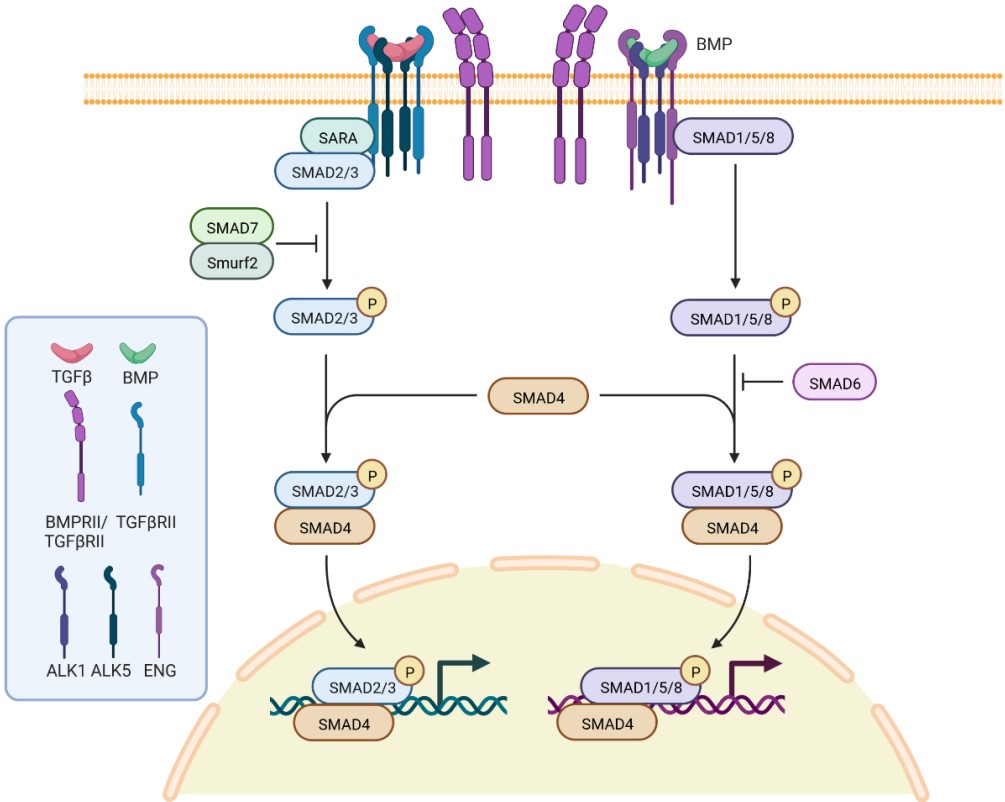

**Figure 1.** Pathway of TGFβ signaling pathway. The image has been obtained through BioRender; Carla Olivieri has been granted a license to use the BioRender content, including icons, templates, and other original artwork under agreement number RE243EDSFS.

## 4. Diagnosis and Clinical Presentation

A clinical diagnosis of HHT is established when at least three out of four of the internationally accepted Curaçao criteria are present in the index case: (i) epistaxis, (ii) mucocutaneous telangiectases in characteristic locations, (iii) visceral lesions, and (iv) a first-degree relative diagnosed with HHT on the basis of the preceding criteria [21]; the diagnosis is "possible or suspected" if only two criteria are present. The efficacy of the Curaçao criteria for the clinical diagnosis of HHT has been confirmed over the years [22,23]. It should be noted that the same authors who proposed the criteria immediately underlined that a more conservative approach should be used in children. In fact, they stated that "if fewer than two criteria are present, HHT is unlikely, although children of affected individuals should be considered at risk in view of age-related penetration in this disorder" [21].

The issue of using the Curaçao criteria in children and teenagers has been addressed in detail by Pahl et al. [22]. They were able to collect retrospective data on about 290 patients, and they correlated the Curaçao criteria with the presence of a pathogenic mutation, taken as the gold standard for the diagnosis of the disease. Based on the evidence that features of HHT are age-related, they also clustered their cases into four different age groups: 0–5, 6–10, 11–15, and 16–21, and data about sensitivity and specificity were obtained for the whole group of cases and for each subgroup. In fact, sensitivity was higher in patients aged 16–21 (91%) than in the 0–5 age group (42%), while specificity was 100% in all groups but the 11–15 one. They concluded that the clinical Curaçao criteria are reliable for the diagnosis of HHT in younger patients also if three or four criteria are present, while in cases where only one or two criteria are recognized, molecular diagnosis will better solve the diagnostic problem.

However, in 2021, Matti et al. considered, in addition to the Curaçao Criteria, nasal endoscopy as a diagnostic tool to improve sensibility and specificity of early diagnosis in pediatric age. Specifically, they conducted a cross-sectional observational study in which

they included 70 children from 47 different families. Each child had a parent with a clinical or genetic diagnosis of HHT. They observed that, "The median age at the onset of epistaxis was 5 years (range, 0.25–15 years), which is earlier than the average of 12 years frequently reported in the literature but in accordance with other recently reported data in pediatric populations". They demonstrated that the integration of nasal endoscopy increased the diagnostic sensitivity of the Curaçao criteria [24].

For clinical symptoms (and diagnostic criteria) such as epistaxis or telangiectasia, the low incidence in lower age groups may hamper the clinical diagnosis both in patients who belong to known HHT families and in patients who will develop the disease because of a de novo mutation.

Often, the diagnosis of HHT is delayed for several years after clinical signs become evident, and this can be at least partially explained by the variable and late penetrance of the different symptoms. However, once the diagnosis of HHT is obtained and confirmed by the demonstration of a pathogenic mutation in the index case, the issue of testing for the presence of potentially life-threatening alteration becomes of high priority and deserves careful attention in all family members, including children.

### 5. Epistaxis (Nosebleeds)

Epistaxes, in general, are very frequent in children, with over 50% of them having the experience of at least one episode, and among them, about 8% requiring treatment (packing, ligation or embolization) [25]. The same authors, while listing the most common causes of epistaxis, do not list HHT, thus confirming both the rarity of the disease and possibly some difficulties in making a clinical diagnosis in children starting from a symptom frequently observed in the general population. In HHT-affected patients, epistaxes are caused by even minor trauma to the nasal mucosa telangiectases, which may be quite numerous; minor trauma may include the simple flow of dry air. Concurrent presence of telangiectases and their very easy bleeding typically results in epistaxis being recurrent, unpredictable, and occurring also at night, when the patient is asleep, and local trauma caused by the child is unlikely. The inclusion of epistaxis among the diagnostic criteria is well supported by the evidence that about 33% of HHT patients experience the nose bleeding onset in childhood, by the age of ten. Their prevalence increases with age, and they are present in 90% of cases before the age of 30 [26]. It is important to note that in many cases, epistaxes are mild, uncommon, do not cause anemia, and do not require medical treatment.

Gonzalez et al. [27] provided a good description of the epistaxis features in HHT children and attempted to define a severity score. The issue of the definition of a severity score has been addressed several times [28–30], as it may be very relevant to evaluating the efficacy of therapies, both in adults and children. As the severity of nosebleeds is usually mild in children, their treatment mainly includes humidification, topical moisturizing, and hemostatic products, while invasive procedures or drugs with possible adverse effects (ablation therapy, septodermoplasty, nasal closure, systemic antifibrinolytic therapy, antiangiogenic agents) are prescribed to adults only. In conclusion, even if HHT is a rare disease, recurrent epistaxes in childhood should prompt all family doctors to collect accurate family histories and eventually to suggest nasal endoscopy to search for telangiectases. It should also be stressed that the latter examination must be performed by an otolaryngologist with extensive knowledge of the disease.

### 6. Telangiectases

Telangiectases are common in the general population, but those associated with HHT are found in specific sites, as the lips, tongue, oral cavity mucosa, fingertips, gastrointestinal (GI) mucosa, and nasal mucosa. Those on the nasal mucosa are obviously present concurrently with epistaxes, while those in other sites appears later, with about one third of cases reporting their appearance before the age of 20 [31]; telangiectases in the GI mucosa, and problems related to their bleeding, are usually a problem of adult life.

Bleeding from telangiectases may be copious and not easy to stop due to the absence of contractile elements in the vessel walls; however, even if the pathological changes are always the same, telangiectases of nasal mucosa bleed more easily and frequently than those of any other site. Telangiectases are unlikely to be a major problem in childhood.

### 7. Arterovenous Malformations (AVM)

Arterovenous malformations (AVM) in the liver, lungs, or central nervous system (CNS) are the most relevant clinical features in adults and represent one of the Curaçao criteria. They may also be present at young ages, without concurrent epistaxis or telangiectases, causing severe clinical complications. Morbidities arise from bleeding or shunting; in some cases, they are life threatening, and their clinical signs may appear suddenly. Therefore, even the occasional observation of the presence of isolated AVMs deserves further investigation, including molecular testing, of the patient and his/her family, to achieve or exclude a diagnosis of HHT. In the case that a diagnosis of HHT is confirmed, a thorough clinical work-up should be offered to all family members, regardless of their age.

The occurrence of AVMs (Figure 2) in pediatric age have mostly been reported as case reports or short series of cases. For instance, Mei-Zahav et al. [32] report the experience of one of the centers with the highest world-wide experience of HHT. In a retrospective analysis of their database, they provide clinical and genetic data on 14 young patients (age 0–15 years) referred to their center, in whom a diagnosis of HHT was suspected but without a known family history of HHT. Among them, 10 out of 14 presented with AVMs: seven in the lung, three in the brain. Noteworthy, in one case, the diagnosis was neonatal, while in most cases the diagnosis of pulmonary AVM (and HHT) was delayed even several years after reporting the initial observation of clinical symptoms, usually cyanosis.

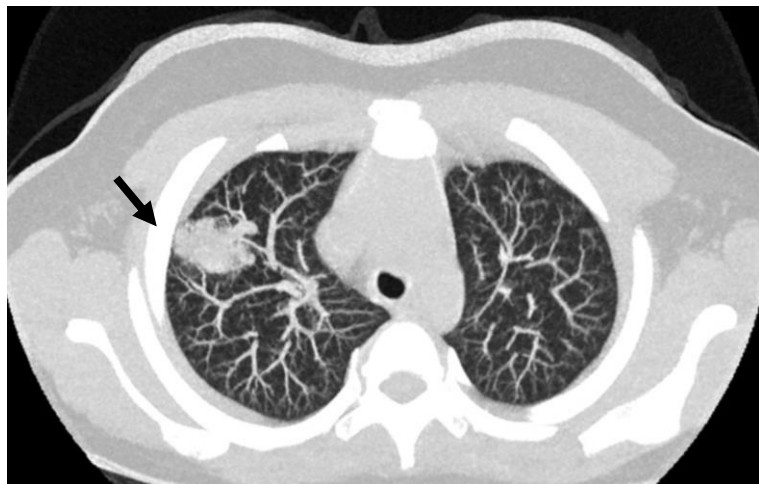

**Figure 2.** Lung CT scan in an 8 years old child; the arrow points to the presence of PAVM in the lung right upper lobe.

In two cases, the diagnosis in the child led to the diagnosis of HHT in an asymptomatic parent, who nevertheless carried pulmonary AVMs, thus strengthening the existence of extended clinical variability and the need for accurate clinical evaluation of all family members at risk of being carriers of the disease-causing mutation.

In the same paper, when the AVM was localized in the brain, the presenting symptoms were seizures or hemiparesis, and the presence of the clinical picture in two cousins allowed for the identification of a large family. In all cases but one, molecular analysis identified either ENG or ACVRL1 mutations.

Krings and coworkers [33], attempted to correlate the different types of CNS AVMs with the patients' age: out of a total of 75 brain lesions, the seven spinal cord AVMs were all observed in young children (mean age 2.2 years), and in 34 AV brain fistulae, 32 cases were

younger than 6 years. They also presented their experience for endovascular treatment in children [34].

The congenital nature of both pulmonary and cerebral AVMs is still a matter of debate [35–37]. If congenital, optimal care for asymptomatic cerebral AVMs remains a dilemma: the issue has been widely discussed and, in our opinion, the paper by Eker et al., 2020 [38] contains a very good analysis of the pros and cons. It seems, nowadays, the most reasonable way to approach the problem. Two of their statements, particularly, deserve to be quoted in this review and used for lightening medical decisions. Statement 2: "The current evidence base does not favour the treatment of unruptured cerebral AVMs, and therefore cannot be used to support widespread screening of asymptomatic HHT patients". Statement 4: "Any screening discussions in asymptomatic individuals should be preceded by informed pre-test review of the latest evidence regarding preventative and therapeutic efficacies of any interventions. The possibility of harm due to detection of, or intervention on, a vascular malformation that would not have necessarily caused any consequence in later life should be stated explicitly".

Joining the data from several authors, and similarly to clinical presentation in adults [39], pulmonary AVMs are more frequent than brain AVMs in children as well, in up to 58% and to 14%, respectively, and in a minority of cases, both occur in the same patient. Of course, the methods used to diagnose the AVMs, for instance, contrast echocardiography (ECHO) grading vs. CT scan in the diagnosis of pulmonary AVMs, may modify the percentage of cases defined as carriers of AVMs. Pulmonary AVMs were reported to have been identified in children with a median age of 9 years if diagnosed following a screening program, and with a median age of 11 years if diagnosed following evaluation of clinical symptoms [32].

The association of a higher risk to develop pulmonary AVMs (as well as epistaxis and telangiectasia) with carriers of ENG mutations was confirmed in children, with figures similar to those reported in the large French-Italian study [39], which included mainly adult cases.

Hepatic AVMs are more frequently associated with mutations in ACVRL1 and have been reported in a wide range of subjects, spanning from 32% to 78% of adults, with only 8% of them showing clinical symptoms. This high variability is at least partially explained with the different methods of ascertainment (sonography or CT scan) [39–41].

In children, although in a very small series of cases (only 19), hepatic AVMs were found in 47% [42], by testing each case with both echo-color Doppler and abdominal CT. In young patients, liver vascular anomalies are classified as vascular tumors (hemangiomas) or vascular malformations with altered flow (low or high). Imaging plays a key role in differentiating them, with relevant implications for diagnosis of HHT and for therapy of the specific vascular lesions.

Relevant cases with clinical evidence of neonatal HHT, also for hepatic AVMs, have been reported [43], although, in general, cases diagnosed in symptomatic children are very rare [44,45]. These reports stress the need for always acquiring a detailed family history, also in cases with apparently isolated AVM in a child. Molecular testing for any young patient presenting with AVM is an option to be taken into account and will be discussed below.

Intestinal AVMs are rare in children, but, in one case at least, intestinal bleeding was evident at birth [46,47].

The improvement of methods to detect prenatal malformations led to the diagnosis of HHT even in utero, although only in exceptional cases; Saleh et al. [48] reported the diagnosis of liver AVM with fetal magnetic resonance imaging (MRI), following ultrasound (US) evidence of cardiomegaly at 32 weeks of pregnancy, and De Luca et al. [49] reported on the presence of a huge malformation of the vein of Galen associated with ventriculomegaly detected with US at 26 weeks, then confirmed with MRI.

In the aftermath of US/MRI diagnosis in both patients, molecular evidence was obtained for mutations in ENG and ACVRL1 respectively. It is regrettable to note that in both cases, familial evidence for HHT was already available [49] or could have been easily

obtained [48] following the Curaçao criteria, but this evidence did not prompt any genetic counselling about the disease in general, nor any advice to the families about the possibility of observing complications of the disease even in children.

In conclusion, considering that AVMs can be observed also in pediatric age, although rarely, and that they are often amenable to therapeutic intervention, offering genetic tests to identify young mutation carriers should be mandatory.

In symptomless children in whom a pathogenic mutation is demonstrated, non- or minimally-invasive tests are available for a front-line evaluation of the patient: oxygen finger saturation, bubble heart ultrasound, neonatal brain ultrasound, and general medical control once a year.

Based on the results of these tests, the appropriate management of the child can be decided: from simply clinical follow-up once a year, to more invasive and frequent tests, up to clinically invasive procedures such as embolization, when needed.

This approach considers both the age of AVM onset and the age-related penetrance variability.

## 8. Heritable Pulmonary Arterial Hypertension and Polymicrogyria in HHT

In addition to the above discussed clinical signs, which overlap with diagnostic criteria, heritable pulmonary arterial hypertension (HPAH) [50], a rare but severe complication of HHT, deserves some comments. The concurrent presence in the same patient of the combination of HHT and HPAH is quite relevant, as it may cause a worse clinical picture of HPAH.

HPAH is recognized by several disease-causing genes, *BMPR2* being the most commonly mutated gene, and is diagnosed at a younger age in *ACVRL1* mutation carriers, compared to *BMPR2* mutation carriers and to idiopathic PAH. In about 30% of cases, HPAH in *ACVRL1* carriers is diagnosed below the age 18 [50] and in some cases as early as 4 years of age [51]. In the last paper, although the above-mentioned Curaçao criteria for the clinical diagnosis of HHT were present in all cases, pulmonary arterial hypertension (PAH) was diagnosed prior to HHT; for all the three reported families, an accurate family history collection allowed the identification of HHT-affected relatives. In one of these families, multiple members have died from PAH at young ages. The association between *ACVRL1* and PAH has been mostly studied starting from a clinical diagnosis of PAH, but Olivieri et al. [52] demonstrated that nine out of sixty-eight (13%) cases selected solely because of the presence of pathogenic *ACVRL1* mutations had increased values of pulmonary artery systolic pressure.

These data clearly suggest that a thorough search for the presence of HHT criteria must be performed in the index cases and in family members of any case of PAH in whom a genetic diagnosis is not yet available, regardless of his/her age. If the Curaçao criteria are recognized in the family, *ACVRL1* should be screened first.

The family described in Figure 3, included among the families reported in [52], depicts very well all the issue to be considered: (i) it is a three-generation family, in which a consistent number of family members carry the pathogenic c.1435 C > T (p. R479X) mutation in *ACVRL1*; (ii) the diagnosis of HHT followed the diagnosis of a very severe PAH (requiring lung transplantation) in IV,9; criteria for the diagnosis of HHT were present in several family members, but underestimated for many years; (iii) IV,9 became pregnant and successfully delivered a male baby; (iv) after genetic counseling, a genetic test in the newborn was performed about 3 days after birth and demonstrated that the mutation was not present. It is obvious how relevant it was to identify the baby as a non-carrier: reassurance of the parents was associated with the identification of the baby as a subject not requiring, at any age, any HHT- or PAH-related clinical investigation.

Alterations of cortical development are new clinical features that have also been recently reported in several cases of both adults and children affected with HHT [53,54]. Klostranec et al. [55] provided a possible explanation to link polymicrogyria and brain AVMs. They suggest that "endoglin mutations, especially those that are dominant-negative, may predispose focal, aberrant hypersprouting angiogenesis during corticogenesis that

leads to the production of polymicrogyria. This hypoxic insult may further serve as the revealing trigger for later development of a spatially coincident bAVM".

In conclusion, even if the first clinical descriptions of HHT are more than 100 years old, evidence is available suggesting that the phenotypic spectrum can be further expanded; thus, reporting unusual findings in HHT patients, even in single cases, is a valuable method to gain further information on the disease phenotype.

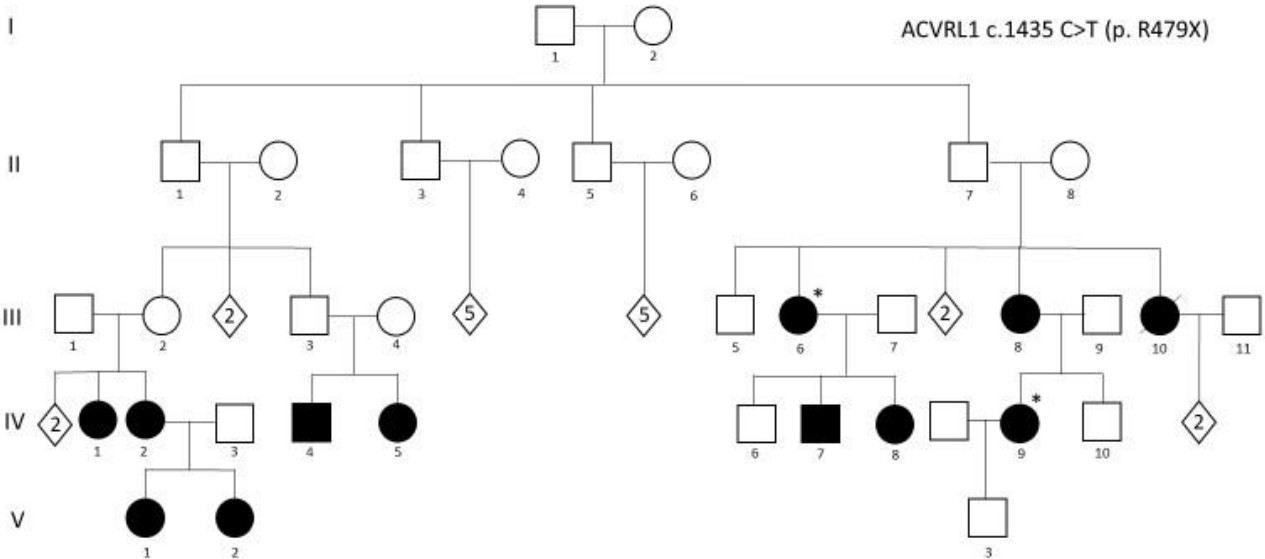

**Figure 3.** Index case IV,9 was diagnosed as affected with pulmonary arterial hypertension (PAH). Black symbols: carrier of ACVRL1 c.1435 C > T (p. R479X) mutation and affected by HHT. "*" indicates mutation carriers presenting both HHT and PAH.

## 9. Pregnancy and Prenatal Diagnosis

To the best of our knowledge, no report has been published suggesting reduced fitness in HHT, and the repeated description of large multigeneration families is in keeping with the presence of normal reproductive skills in HHT patients. Most pregnancies in women affected by HHT have a normal course and outcome; for instance, miscarriage and prematurity are observed as frequently as in the general population [56]. Maternal risks for life-threatening events, such as major PAVM bleed, stroke and death, were estimated to be about 1% for each complication by Shovlin et al. [57] and will result in fetal risks for intra-uterine death or intra-uterine growth retardation, secondary to chronic hypoxemia, maternal hypovolemia or maternal death [56]. Of course, the occurrence of this severe complication can be greatly reduced by appropriate pregnancy management: HHT women planning a pregnancy should be screened and treated for pulmonary and cerebral AVMs prior to conception. Pulmonary AVMs discovered during pregnancy can be treated in the second trimester. A consensus has been reached stating that screening for spinal vascular malformations is not required, and that there is no reason to withhold an epidural in HHT pregnant ladies [23].

Prenatal presentation of HHT is possible in cases of severe anatomical abnormalities [48,49], so, in general, any pregnancy in which the fetus is at risk for having inherited the disease should be accurately monitored with repeated US examinations. Our group was involved in genetic testing (unpublished data) after abortion of a fetus because of severe cerebral hemorrhage at the 18th week; no autopsy was performed, but on a sample of fetal tissue, the familial mutation was confirmed.

Of course, if the familial pathogenic mutation is known, prenatal molecular testing becomes feasible using any fetal cell type, with chorionic villi or amniotic fluid cells as the most common. The recent advances in non-invasive prenatal tests for monogenic disorders, discloses this possibility also for HHT [58].

Prenatal genetic testing should be extensively discussed during genetic counseling; while the identification of the pathogenic mutation causing the disease in the family can be easily obtained, conversely, the demonstration of the mutation in fetal cells will not give any information useful to forecast disease course or severity.

One pro for prenatal testing is definitely the possibility of reassuring parents if the mutation is absent, but a relevant cons are represented by the risk of creating unmotivated anxiety. In addition, a genetic test yielding the same information can be quickly obtained at birth. Certainly, a prenatal molecular test must not be performed without clear parental comprehension of all its consequences, including fetal risks associated with chorionic villi sampling.

## 10. Genetic Counseling

As for any other genetic disorder, genetic counseling is a fundamental part of disease management, and it is appropriate to offer it to any patient and, in particular, to young adults who are affected or at risk. It must be given by a specifically trained counselor, who will discuss all the HHT-related issues, such as: (i) risk evaluation for being carrier of a pathogenic mutation for all close relatives of an index case; (ii) genetic and clinical heterogeneity; (iii) concerns of genetic testing, family planning and prenatal diagnosis. The counselor also needs to have a good knowledge of the disease, to integrate genetic and clinical data in a general overview, particularly in young ages, when application of clinical diagnostic criteria may be unsuccessful to identify affected children [21,22].

Genetic counseling is of the greatest importance in explaining molecular analyses results to the families: clinical significance of a pathogenic mutation and its impact on the family; clinical and genetic significance of a negative result (no mutation found); how to manage the presence of VUS (variants of unknown significance).

## 11. Guidelines

Faughnan et al., in 2020 [23] published the Second International Guidelines for the Diagnosis and Management of Hereditary Hemorrhagic Telangiectasia and approved six recommendations, which highlight new evidence in existing topics from the first International HHT Guidelines. They also provided guidance in three new areas: anemia, pediatrics, and pregnancy and delivery. The topics addressed for pediatrics include molecular diagnosis, and screening for pulmonary and brain AVMs.

Testing asymptomatic children of an HHT parent was highly recommended, with 94% agreement. In fact, there is good evidence for the relevance of testing in identifying subclinical or presymptomatic disease in children [43].

Screening asymptomatic children "with HHT or at risk for HHT" for pulmonary or brain AVMs at the time of presentation/diagnosis yielded a strong recommendation, but the agreement for screening for pulmonary AVMs was much higher than for brain AVMs (94% vs. 86%). There is a large agreement (98%) for treatment of large pulmonary AVMs associated with oxygen saturation reduction, and the same is true for brain AVMs if high-risk features are present. As expected, the relevance of genetic counselling to help decision processing is stressed.

It is underlined how the medical attitude about screening for brain AVMs may vary greatly in different countries, and how the patients' representatives who collaborated in the drafting of these guidelines were in favor of screening.

## 12. Molecular Testing

Genetic testing in children has been a matter of debate for a long time, and the British Society for Human Genetics [59], in 2010, published new guidelines on this topic. According to these guidelines, genetic testing should be offered "when a child is at risk of a genetic condition for which preventive or other therapeutic measures are available".

In addition, genetic testing should also be considered if necessary to prevent serious harm to other family members.

Children at risk for HHT, based on available data on the disease, are in fact at risk for several clinical problems for which therapeutic interventions are (partially) available; thus, there is a good indication for genetic testing.

Several methods can be used to identify a pathogenic mutation, and both clinical and family history will suggest the best choice, always after appropriate genetic counseling.

The easiest scenario is, of course, the one in which a child belongs to a family with a known proven diagnosis of HHT and in which a pathogenic mutation is known. Given the autosomal dominant mode of inheritance of HHT, any child born to a clinically affected parent in whom a pathogenic mutation has been identified has a 50% chance of carrying the same mutation. We may consider two different circumstances: (i) The child is completely symptomless, as it may be in newborns. Clinical surveillance is indicated; invasive tests should be avoided. Genetic testing is certainly needed and should be performed after extensive genetic counselling with the family, discussing the pros and cons of testing asymptomatic children. Moreover, "labelling" an asymptomatic person as a mutation carrier and an HHT patient in an unpredictable future is an issue to be discussed. (ii) The child shows any HHT-related symptom (even mild epistaxis). Testing should be offered and performed as soon as possible. In both cases, the indication is to test only for the familial mutation.

A second scenario is when the child shows HHT-relatable clinical signs, but no clinical and molecular diagnosis have already been obtained in the family, in which, however, some other members show clinical symptom. In this case, an accurate first-degree relative's workup should be performed, first testing the subject best fitting the HHT diagnosis.

Finally, in the third scenario, the child shows a clinical picture suggestive of HHT in the presence of a completely negative family history. Of course, occasional causes of sporadic mild epistaxis should have previously been ruled out.

In the latter two cases, an old approach is the serial single-gene testing, performing sequence analysis of ACVRL1 and ENG first, followed by deletion/duplication analysis of the same genes if no pathogenic variant is found. Analysis of SMAD4 becomes the first choice if any evidence of association with HHT and intestinal polyps is available. Additional genes (i.e., GDF2, RASA1, and EPHB4) will be analyzed if no mutation is found. However, this method is time-consuming and expensive and is nowadays outdated. Better options are: (i) Testing the index case with an NGS multigene panel specifically designed for HHT, including all the above-mentioned genes. The main limitation is that the number of included genes, and thus the panels' sensitivity, may vary in different laboratories; furthermore, all multigene panels designed for a specific disease will change over time. (ii) Whole exome sequencing followed by in silico analysis of the genes of interest. The great advantage of the latter is that in silico analyses can be repeated, even after years, without requiring a new blood sample from the patient, if a mutation has not been found after the first results analysis or if a novel clinical diagnosis suggests the involvement of different genes.

Again, the results of genetic testing must always be given to the families within a genetic counseling session (see above).

The laboratories testing HHT-related genes with any method must, of course, have all the expertise to decide how and when further genetic analyses (for instance, targeted deletion/duplication analysis) are indicated.

If a pathogenic mutation is found in a child who is the first member of the family to be tested, after appropriate genetic counseling, mutation analysis should be extended to parents first, and then to all other family members at risk of having inherited it.

## 13. Surveillance

Annual follow up by a general practitioner (familiar with HHT) could be appropriate, considering the specific clinical picture of the patient.

Epistaxes may recur after treatment also, and nasal endoscopy in children can be repeated without any major problem in order to assess if any further treatment is indicated [24].

Pulmonary AVMs may become evident or increase with time (see Section 14 for reference). Therefore, noninvasive screening (i.e., using transthoracic contrast echocardiography or chest radiography associated with pulse oximetry) is recommended in asymptomatic or at-risk children. When a reduced oxygen saturation and/or a feeding artery larger than 3 mm are found, pulmonary AVM treatment is recommended. A five-year surveillance interval is suggested for negative subjects, while differences among centers are known regarding protocols and timing for positive or post-treatment children's follow up [23].

Screening for brain AVMs is discussed in Section 11. If performed in infancy, control after puberty is suggested [7].

In families in which a diagnosis of *SMAD4*-HHT is strongly suspected or proven with molecular testing, colonoscopy should be performed first at age 15 [7] and repeated at three years' intervals in the absence of any polyp. If polyps are present in the colon, surveillance will be performed annually with esophagogastroduodenoscopy.

## 14. Therapy

Therapy addresses the relevant clinical issues in each child and thus depends on clinical presentation. Several therapeutic protocols have been proposed in adults (including the use of thalidomide and bevacizumab) to control abnormal vessel development. Although some encouraging results have been reported, their use is limited to selected adult cases with severe epistaxis [60–62].

All other therapies available for adults can be offered also to children, after accurate clinical evaluation of the patient. For instance, the youngest patient undergoing argon plasma coagulation was 6 years old in the series of cases reported by Khan et. al. [63].

Treatment of brain AVMs may include embolization, stereotactic radiosurgery, and surgery, and can be applied to children as young as 1 year of age [64]. The results are encouraging as, after long term follow up, over 90% of children have been reported to show a good outcome [64].

Pulmonary AVMs in children, as in adults, can similarly be treated by embolization [65]. It is noteworthy, in this paper, that there was an observation of some cases with increasing size of the lesion with time or its appearance after a first negative screening. These data highlight the need for continued surveillance in order to evaluate if a repeated therapy is necessary.

**Author Contributions:** Conceptualization, C.D. and C.O.; writing—original draft preparation, C.D.; writing—review and editing, C.O. and C.C. All authors have read and agreed to the published version of the manuscript.

**Funding:** This research received no external funding.

**Institutional Review Board Statement:** Not applicable.

**Informed Consent Statement:** Not applicable.

**Data Availability Statement:** Not applicable.

**Acknowledgments:** We thank HHT Patients and the two HHT Italian Charities, Associazione Fondazione Italiana HHT "Onilde Carini" and HHT Onlus, for their continuous support to our studies.

**Conflicts of Interest:** The authors declare no conflict of interest.

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
