# Peer review of "Hereditary Hemorrhagic Telangiectasia in Pediatric Age: Focus on Genetics and Diagnosis"

_pediatrrep, doi:10.3390/pediatric15010011_

Round 1

Reviewer 1 Report

see in attachment

Reviewer 2 Report

This manuscript provides the overview of hereditary hemoeehagic telangiectasia (HHT) in pediatric population. Because HHT is rare disease and the children with HHT often exhibit few signs and symptoms of the disease, the diagnosis and managemant of the affected children differ from those of adult patients. This review  is importanto for pediatrcians .

But I want to ask the authors about the guideline of the management of pediatric patients with HHT. Although the authors referred the 2nd international guideline of HHT, they did not mentioned about the details concernig about pediatric patients with HHT or diagonosis of asymptomatic young chidren described in the guideline.

 Please describe the pediatric care of HHT and genetic testing for asymnptomatic children of a parent with HHT according to the guidline. These would be a standards for the management of the children with HHT.

Reviewer 3 Report

Authors have done a comprehensive review of HHT in the pediatric age group.

Introduction: Please add reference for line 44, 45.

I dont think Figure 1 is of much value. 

Diagnosis: Line98: Please add reference for efficacy for the diagnostic criteria.

Line 110 to 111: Considering rephrasing for better understanding

Guidelines: It would be nice to discuss more about the guideline. 

Surveillance: Please elaborate

Need to expand on therapies more. Last statement is backed up by evidence or authors are extrapolating?

Round 2

Reviewer 1 Report

I thank the Authors for performing the required modifications in the manuscript. I accept it in the present form.

Author Response

We wish to thank Reviewer 1 for the positive comments on our reviewed version of the manuscript.

Reviewer 3 Report

Regarding fig 1, I agree with the point of giving credits but this is a scientific review and reference to prior paper and appropriate inclusion in discussion would suffice. 

Line 480: pulse ox may be used as a screening test, but will that exclude? Is it provided in the reference paper. ?

Line 481 to 484: please rephrase it for better flow of information.

Line 499 - 500: Authors referenced that the cited paper discussed about results but results were not discussed in the therapy section. Please redo it. 

Line 503: rephrase to "highlights the need for continued  surveillance"
